# OpenReview forum: "The Geometry of Phase Transitions in Diffusion Models: Tubular Neighbourhoods and Singularities"
_TMLR — Accepted by TMLR_

### Review · Reviewer_Mx5F · 2025-05-26

**Summary Of Contributions:**

The authors provide a novel perspective on phase transitions in diffusion models. In particular, they analyze them from a geometrical perspective. Leveraging tools from tubular neighbourhood theory, they can define regions where phase transitions occur, which is validated by empirical results.

**Audience:**

Yes

**Broader Impact Concerns:**

No broader impact concerns.

**Claims And Evidence:**

Yes

**Requested Changes:**

1) Especially in Section 4.2, the analysis gets very dense. The authors should try to comment more on their results. Stating beforehand what we will prove now and why and afterward commenting on the implications of the results.

2) In Section 2.2, the authors simply state that their paper is based on the hypothesis that data often concentrates around a lower-dimensional manifold. Could you comment on the implications of this assumption? Also on the one following, that you assume all data manifolds are compact and embedded in the Euclidean space?

3) Could you elaborate a bit more on Remark 4.6? In what sense is it a generalisation, and what exactly is the result that you are citing?

4) There are a few language issues/typos. Between (2) and (3), there seems to be an article missing before "probability distribution." Also, the "we could" reads a bit strange, maybe that could just be deleted. In the second sentence of Section 3, there is an "in" too much. Before Theorem 6, I think the "come into" is too much.

**Strengths And Weaknesses:**

Strengths

The paper appears to provide a novel perspective with potential for increasing sampling efficiency in diffusion models. The theoretical analysis seems thorough, and the empirical study confirms its results.

Weaknesses

The mathematical analysis is very dense in parts. Adding some comments to set the theoretical results into perspective would help the paper be more accessible. Also, assumptions should be discussed a bit more. Overall, the language is good, and the paper is relatively easy to follow. I comment on a few issues below.

---

> ### Author Response · Authors · 2025-06-20
> **Response to Reviewer Mx5F — Requested Changes 1–4**
>
> ## Requested Changes 1
>
> Thank you for flagging the readability of § 4.2.
> While we have carefully double-checked that the original proofs are correct, we recognise that their highly formal presentation may have felt imposing. To make the section more accessible—especially for a broader audience beyond core mathematicians—we have introduced several light-weight guides:
>
> 1. Preview sentences at the start of each lemma/proposition that state *what* is being proved and *why it matters* in the overall argument.
> 2. Brief concluding remarks after each proof that spell out the key implication in plain language.
> These additions preserve full rigour while lowering the entry barrier. We hope the revised draft reads more smoothly and would welcome any further suggestions. Thank you again for the helpful feedback.
>
> ---
>
> ## Requested Changes 2
>
> Thank you for raising this important point regarding the assumptions that underpin our work.
> The *manifold hypothesis*—that high-dimensional data concentrate near a substantially lower-dimensional manifold—has become a standard premise in modern machine-learning theory. Empirical support spans early nonlinear dimensional-reduction studies (e.g., LLE and Isomap) to more recent large-scale studies of image, all of which consistently reveal low intrinsic dimensionality. Our reliance on this hypothesis is therefore motivated by accumulated evidence rather than methodological convenience.
> Building on this empirical premise, we additionally assume that the data manifold is smooth, compact, and smoothly embedded in the ambient Euclidean space $\mathbb{R}^{d}$. Once one accepts a smooth data manifold, classical tools from differential geometry, in particular *tubular neighbourhoods*, become available.
> We have incorporated this clarification into Section 2.2 and sincerely thank you for bringing these important points to our attention.
>
> ---
>
> ## Requested Changes 3
>
> Thank you for asking for more detail. Prior work (Stanczuk et al., 2024, Theorem D.1) shows that in the limit of an infinitesimal step size $t \to 0$ the score vector aligns with the *normal* direction.
> Our remark broadens this statement: **whenever a specific inequality relating the step size, the ambient dimension, and the injectivity radius is satisfied, the score vector instead points toward the manifold side.** Because the inequality automatically holds when the step size is sufficiently small, our claim reduces to Theorem D.1 as a special case. We have incorporated this explanation into Remark 4.6 for ease of reference.
>
> ---
>
> ##  Requested Changes 4
>
> Thank you for spotting these slips. We have carefully revised the manuscript (and the mirror passage in **Appendix G**) as follows:
>
> | Location | Revision |
> | --- | --- |
> | **Sec. 2.1, lines 73–74** | Removed the phrase *“we can derive”*. |
> | **Appendix G, line 977** | Updated wording for consistency. |
> | **Before Theorem 3.6, line 134** | Re-phrased as *“… at which two separated tubes **touch each other**.”* (removed *come into*). |
>
>
> In addition, we ran a full proof-reading pass (Grammarly + human) to eliminate similar minor issues throughout the paper.
>
> We hope these edits resolve the reviewer’s concerns.

---

> > ### Comment · Reviewer_Mx5F · 2025-06-23
> >
> > I think the changes helped clarify the paper. When going through the changed parts, I noticed a few minor details. In Sec. 2.1, in the first sentence, the reference to Hyvarinen et al. should be in parentheses. In Remark 4.1, there is a space missing. The sentence before Equation (8) is a bit unclear; I'm not sure whether something is missing or whether the second abbreviation should be without parentheses. In Remark 4.8, I believe the first sentence is missing an article, and I'm unsure what Section I.2 is referring to. If it corresponds to the section in the appendix, I would recommend making that explicit.

---

> > > ### Author Response · Authors · 2025-06-23
> > > **Response to Reviewer Mx5F**
> > >
> > > Dear Reviewer Mx5F,
> > >
> > > Thank you for your careful reading and helpful remarks.
> > > We have incorporated all of your suggestions in the revised manuscript now uploaded:
> > >
> > > - **Line 65** – The citation to *Hyvärinen et al.* (2005) is now parenthetical.
> > > - **Line 180** – Added the missing space before “(see also …)”.
> > > - **Lines 191–192** – Merged the consecutive abbreviations into “(VP-SDE, also known as DDPM)” for clarity.
> > > - **Line 217** – Added the missing article **“a”** and changed “Section I.2” to **“Appendix I.2”** to make the reference explicit.
> > >
> > > We will continue to proof-read for any remaining minor issues as time permits.
> > > Thank you again for helping us improve the paper.
> > >
> > > Best regards,
> > > *The Authors*

---

### Review · Reviewer_L9AT · 2025-06-08

**Summary Of Contributions:**

The paper presents a new concept known as the injectivity radius to explain the emergence of phase transition phenomena in diffusion models. The core intuition is that once the diffusion process enters a certain so called tubular neighborhood, where projection to the manifold is uniquely determined, generating data that conforms to the structure of the manifold becomes easier, and that the boundary of such tubular neighborhoods help understand phase transition. The authors propose an algorithm to estimate the injectivity radius and empirical evidence to support the theory.

**Audience:**

Yes

**Broader Impact Concerns:**

No ethical concern.

**Claims And Evidence:**

Yes

**Requested Changes:**

- correct any typos and formatting issues.

**Strengths And Weaknesses:**

Strengths:
- Idea: The idea of injectivity radius and tubular neighborhoods for understanding diffusion model phase transitions is highly creative and highly original
- Clarity: The theoretical parts of the paper, even though they involve more advanced mathematical concepts/new definitions, are generally clearly written. The motivation of the paper is also presented clearly.
- Evidence: The experimental evidence, while focusing on relatively simple cases, covers enough ground to illustrate the plausibility of the theory.

Weaknesses:

1. There are a number of spelling mistakes, capitalization/other formatting issues in the appendix of the paper. I urge the authors to take a detailed pass to fix these grammatical/spelling issues.

2. This is not really a weakness, but I still would like to point out a limitation of the study. Basically, the authors introduced the notion of tubular neighborhoods and injectivity radius, and showed some theory that the probability moves towards the manifold. There is actually little theory that points towards a phase transition (if I am incorrect in my understanding I welcome any corrections). The evidence towards the relevance of the injectivity radisu towards phase transition come purely from the experiments. It would be nice to see some theory that actually links the injectivity radius to phase transitions. However, the current material in the paper more than suffice for TMLR, and I am NOT requiring the author to make any edits/do anything about this. I am just suggesting a future direction.

---

> ### Author Response · Authors · 2025-06-20
> **Response to Reviewer L9AT — Weaknesses 1–2 Addressed**
>
> ## Weakness 1
> Thank you for catching these oversights. We have now performed a careful, line-by-line proof-reading of all appendices, corrected the identified spelling and capitalization errors, and harmonised the formatting. A fully revised manuscript has been uploaded. We appreciate your vigilance in helping us improve the presentation.
>
> ---
>
> ## Weakness 2
> Thank you for raising these essential points. As you note, our work introduces the injectivity radius and provides a rigorous theorem on the score-vector field. We see this as an important first step toward giving a geometric perspective on phase transitions. At the same time, as Section 4.2 makes clear, the injectivity radius supplies only a *sufficient* condition for the time at which the score vector flips orientation; it is not yet a necessary one, leaving room for a more precise characterisation of the phenomenon.
>
>
> To motivate future work, Section 6 outlines several avenues for improvement. In particular,
>
>
> * the current injectivity radius is governed by the largest principal curvature; analysing an analogous mathematical quantity tied to the smallest curvature may reveal additional critical behaviour, and
> * constructing a system-level free energy by weighting the energy at each point with the probability distribution function could possibly provide a thermodynamic lens through which phase transitions can be detected more accurately.
>
>
> We hope this discussion clarifies the scope of our contribution and highlights promising directions for a sharper theoretical treatment. Once again, thank you for stimulating this important conversation.

---

### Review · Reviewer_RQ3i · 2025-06-08

**Summary Of Contributions:**

The goal of this paper is to offer a new and different perspective on phase transition observed in previous work during the generative process of diffusion models. To this end, the authors borrow some ideas and theorems from the manifold analysis and differential geometry literature, notably, tubular neighborhoods and injectivity radius. They propose an algorithm to estimate the injectivity radius of data manifold, which they employ on simple synthetic data as well as real data. They demonstrate empirically that phase transition happens around the borders of Tubular neighborhoods.

**Audience:**

Yes

**Broader Impact Concerns:**

In my opinion the broader impact of this type of work which aims at shedding light on understanding deep neural network is positive in the long run.

**Claims And Evidence:**

Yes

**Requested Changes:**

**Necessary adjustments:**
- An important condition required for the analysis is the smoothness of the manifold (also finite curvature). For clarity, it’s best that these assumptions are stated early on in the text. This is specially important in the context of images, since there is both empirical evidence and theoretical arguments that these assumptions do not hold for image distributions.
- Beyond simple analytic manifold, how does one measure the infectivity radius for a data manifold? It is not clear if that is possible for real world data manifolds. But having access to this value is required to then use the analysis in the paper. One solution to this problem, implemented by the authors, is to embed the image manifold into a known simpler manifold as a preprocessing step. But the problem with this approach is that then the resulting analysis does not tell us much about the actual manifold and trajectory towards it but an $S^{20}$ manifold. (And it is not clear if even such trick would hold beyond MNIST)

- The link between the injectivity radius and Wasserstein distance is not made clear in the text, resulting in a disconnect between the theory and experiment parts. The relationship can be made more explicitly discussed. Why is it a relevant measuremtment?

- Generally my main criticism of the work is that although using the idea of injectivity radius is an innovative and geometric way to explore the phase transition phenomenon in diffusion generation, and the authors successfully capture the relationship between injectivity radius and phase transition in smooth simple manifolds, the paper is not convincing that this analysis would hold in the case of the image manifolds. So my proposal is to edit the text to reflect the scope and limitations of the analysis

 **Recommended adjustments:**

- A more explicit and in depth discussion of phase transition phenomenon in an introduction or background section will improve the motivation and analysis proposed in the paper. (This is as opposed to only citing the papers which introduce this observation). Since the entire paper is built on the assumption that these abrupt phase transitions exits and are detectable, it seems necessary to first establish them for the reader. (Again this is important for manifolds arising from real work data beyond simple manufactured manifolds).
- In section 2.2, the manifold hypothesis is attributed to more recent work. However, this hypothesis was introduced earlier:
  - Roweis, S.T. and Saul, L.K., 2000. Nonlinear dimensionality reduction by locally linear embedding. science, 290(5500)
  - Tenenbaum, J.B., Silva, V.D. and Langford, J.C., 2000. A global geometric framework for nonlinear dimensionality reduction. science, 290(5500)
  - Probably more original paper which can be found by a quick search
- In the experiment section, the more interesting manifolds are the Ellipse and the Torus, because in these cases the injectivity radius can be manipulated by changing the major and minor axes. The hypersphere cases do not carry much information in my opinion, so they can be removed from the text. Instead you can show how the tubular neighborhood changes in an Ellipse with different major to minor axes ratio (which can be more revealing of the theory in the earlier part).

**Strengths And Weaknesses:**

**Strengths:**

- The idea of using Tubular neighborhoods for describing phase transition (feature emergence) is creative and to my knowledge is novel. It provides a nice geometric perspective on emergence of features which is linked to phase transition.

- The writing and the back ground is generally clear although it can be improved.

**Weaknesses:**
- Limitations and assumptions of the theory can be more clearly delineated (see next section)

---

> ### Author Response · Authors · 2025-06-20
> **Response to Reviewer RQ3i — Necessary Adjustments 1–4 (Assumptions & Scope Clarified)**
>
> ## Necessary adjustments1
>
> Thank you for pointing this out; we agree that clearly stating the assumption of our geometric analysis is essential.
>
> ### What we changed
> * **Section 2.2 has been rewritten.**  It now contains
>   * **Assumption 2.1** (“Smooth manifold with bounded curvature”),
>   * a new *Why embrace the hypothesis?* paragraph motivating the assumption,
>   * a *Scope.* paragraph that delimitates when the analysis can be trusted, and
> * **Section 6, second paragraph** now reiterates that real-world image manifolds may violate these regularity conditions.
>
> Please see lines 77-103 and 348-352 of the revised PDF for the exact edits.
>
> ### Why this addresses your concern
> The new text makes it explicit that
> 1. all theoretical guarantees rely on Assumption 2.1;
> 2. the assumption holds for our synthetic experiments but can fail for natural images, in which case our results should be interpreted qualitatively; and
> 3. extending the analysis to stratified or highly curved spaces is left for future work.
>
> We hope these edits satisfactorily clarify the intended scope of our contribution and thank you again for the insightful feedback.
>
> ---
> ## Necessary adjustments2
>
> Thank you for raising this important point.
> We address it from both a **theoretical** and an **experimental** perspective.
>
> ### (A) Clarifying the scope in the main text
> * **Section 2.2 – Scope.** now states explicitly that our geometric machinery requires a compact smooth submanifold of $\mathbb{R}^d$.
>
>   Data that instead form a *stratified set* (with corners, edges, etc.) would necessitate a “tube system” whose projection is piece-wise and numerically unstable, making them unsuitable for practical computation.
> * **Section 6, second paragraph** reiterates this limitation.
>
> ### (B) Operational workaround on benchmark datasets and its limits
> Your observation is correct in that we do not measure the true injectivity radius of the raw image manifold.
> Because such a value is generally inaccessible for real-world data, we adopt an *operational workaround* in our experiments:
>
> 1. **Hyperspherical embedding.** We map the dataset onto a hypersphere via a Spherical VAE (SVAE).
> 2. **Known radius on the sphere.** On the sphere the injectivity radius is determined by its curvature, allowing us to gauge how late initialisation relative to this scale influences sampling dynamics.
>
> The same SVAE-based embedding can in principle be extended to higher-resolution datasets such as CIFAR-10 or ImageNet.
> However, as noted in Appendix E of the SVAE paper, the surface area of the unit sphere $\mathbb{S}^n$ decreases once $n \ge 8$ [1].
> For latent spaces of higher dimension one must **tune the radius of the target hypersphere** so that its surface area matches the intrinsic spread of the dataset.
> Unlike a Gaussian prior in an ordinary VAE, the hyperspherical prior is uniform, making the radius the sole free parameter; devising dataset-specific schedules for this parameter is a non-trivial engineering task and is therefore listed as future work at the end of Section 6.
>
>
> We hope these additions clarify how the injectivity radius is handled in practice and delineate the boundaries of our current contribution.  Thank you again for the constructive feedback.
>
> [1] Davidson, T. R., Falorsi, L., De Cao, N., Kipf, T., & Tomczak, J. M. (2018). Hyperspherical variational auto-encoders. arXiv preprint arXiv:1804.00891. Appendix E.
>
> ---
>
> ## Necessary adjustments3
>
> Thank you for pointing this out.
> Our intention is *not* to present 1-Wasserstein distance as a theoretically bounded error metric, but rather as a qualitative, parameter-free indicator of how the generated distribution changes under *late initialisation*. We have updated the manuscript to make this role explicit (line 248-249).
>
> ---
>
> ## Necessary adjustments4
>
> Thank you again for raising this central issue.
>
> ### What we changed
> * **Section 2.2** now opens with **Assumption 2.1** and adds a dedicated *Scope.* paragraph that states, without qualification, that our theory applies only to data supports satisfying this assumption or to datasets that can be smoothly embedded into such a manifold.
> * **Section 6, second paragraph** summarises the same limitation in plain language.
>
> ### Why a full extension to image manifolds is non-trivial
> The injectivity radius of a realistic image manifold cannot—at present—be computed analytically.
> When the support possesses sharp edges, occlusions, or self-intersections, it is better modelled as a *stratified set*.  Although a so-called “tube system” can be defined for such sets, its normal-projection map is only piecewise injective and becomes numerically unwieldy, making a rigorous bound on phase-transition timing out of reach.
>
>
> We hope these additions make the intended scope of our analysis unmistakable while indicating a clear path for extending the theory to real-world data.  Thank you again for helping us refine the presentation.

---

> > ### Author Response · Authors · 2025-06-20
> > **Response to Reviewer RQ3i — Recommended Adjustments 1–3**
> >
> > ## Recommended adjustments1
> >
> > We appreciate the reviewer’s suggestion to give a more extensive account of abrupt phase transitions.
> > Our results indeed rest on the empirical observation that if the initialization of Gaussian noise is delayed, the dynamics never experience the transitional regime that would normally unfold inside the injectivity-radius scale; skipping that regime produces a pronounced shift in the eventual log-density of the learned distribution. A complete theoretical picture of this effect would certainly strengthen both the paper and the broader literature.
> >
> > As noted in Appendix G and echoed in § 6 (Discussion & Conclusion), a rigorous theoretical explanation of this transition is still an open, technically demanding problem. Exploring it in depth would require a survey of geometric measure theory, stochastic thermodynamics, and recent numerical evidence—material that lies beyond our page budget and could distract from the paper’s central contribution.
> >
> > For these reasons, we have chosen not to expand the main text further and instead direct interested readers to Appendix G for background and open questions. We hope the reviewer will agree that this strikes an appropriate balance between completeness and focus.
> >
> > ---
> >
> > ## Recommended adjustments2
> > Thank you for catching this oversight.
> > We have now added an explicit sentence at the start of § 2.2 acknowledging the original formulation of the manifold hypothesis and citing Roweis & Saul (2000) and Tenenbaum et al. (2000) accordingly.
> >
> > ---
> >
> > ## Recommended adjustments3
> >
> > Thank you for the suggestion. As you point out, the Ellipse and Torus experiments yield particularly interesting results, and we find them compelling as well. However, we present the hypersphere first so that readers can build intuition step-by-step on the simplest geometry before moving on to the Ellipse and Torus. We then discuss the Ellipse and Torus in turn. Variants with different axis ratios (Ellipse) and radii (Torus) are provided in the Appendix. We hope that this progression helps readers deepen their understanding. Thank you again for the helpful proposal.

---

> > ### Comment · Reviewer_RQ3i · 2025-06-25
> >
> > Thanks you for your response. The edits have improved and clarified the paper.

---

### Author Response · Authors · 2025-06-20
**Major Update: Theory Assumptions Clarified, Manuscript Polished**

Dear Reviewers,

Thank you very much for your thoughtful and constructive comments.
Our study combines **novel mathematical ideas** with experiments to advance the understanding of phase-transition phenomena, yet we recognise that the original manuscript did not state our theoretical assumptions clearly enough. To clarify these **assumptions** and their **scope**, we have added new text and revised several sections. Below we summarise the main changes, each of which responds to points raised by one or more reviewers.


### Section 2.2
  1. State the regularity assumptions (compact smooth submanifold) and the rationale behind them.
  2. We describe the scope of applicability of our method and outline the challenges that arise when extending it to cases such as image manifolds.

### Section 6
We restate the assumptions underlying our theory of the injectivity radius and highlight the remaining challenges.

### Editorial improvements
We performed a careful edit of the entire manuscript (especially the Appendix) to correct typographical errors and improve overall readability.

We believe these changes make the paper’s contributions clearer and more solidly grounded.
Once again, thank you for helping us improve the manuscript.

Sincerely,
*The Authors*

---

### Decision · Action_Editor_Hmr2 · 2025-07-07

**Recommendation:** Accept as is

**Additional Comments:**

I recommend the **Featured Certification** for this paper. It introduces a rigorous and original theoretical framework using injectivity radius and tubular neighborhoods to explain phase transitions in diffusion models. The interdisciplinary approach, drawing from differential geometry and generative modeling, is both creative and well-executed. The work stands out for its theoretical depth, clarity, and potential to influence future research in this area.

**Audience:**

Yes

**Audience Explanation:**

This submission aligns well with the TMLR readership, particularly those interested in **generative modeling**, **geometric deep learning**, and **mathematical foundations of diffusion processes**. While the concepts may be mathematically dense, the clarity of writing and careful response to reviewer concerns make the content accessible to the intended audience.

The perspective is most relevant for **researchers exploring theoretical underpinnings of generative models** and the geometric nature of their latent spaces. Due to the reliance on assumptions that may not hold in practice (e.g., real image manifolds), the contribution may be more appealing to **theoretical oriented readers**.

**Claims And Evidence:**

Yes

**Claims Explanation:**

This paper offers a novel geometric perspective on phase transitions in diffusion models by leveraging concepts such as **tubular neighborhoods** and the **injectivity radius**. The authors propose that phase transitions arise when diffusion trajectories exit the injectivity radius around a data manifold and demonstrate this hypothesis with both theoretical development and empirical evidence on synthetic and real data. The concept of using geometric constructs to explain abrupt feature emergence adds a fresh viewpoint to the literature.

The reviewers found the paper **original and technically sound**. They highlighted that the theory is rigorous and the experimental results, while based on relatively simple manifolds, offer plausible support for the claims. The connection between manifold geometry and diffusion model behavior is presented with clarity, and the authors responded constructively to comments, refining assumptions, scope, and presentation throughout the paper.

While some theoretical links, especially between the injectivity radius and phase transitions, remain largely empirical, the work succeeds in laying a foundation for further development. Assumptions about manifold smoothness and compactness are now clearly stated, and limitations in extending the theory to complex real-world data (like natural images) are acknowledged.